# Density-aware Chamfer Distance as a Comprehensive Metric for Point Cloud Completion

**Tong Wu**[1], **Liang Pan**[2], **Junzhe Zhang**[2,4], **Tai Wang**[1,3], **Ziwei Liu**[2], **Dahua Lin**[1,3,5]
[1]SenseTime-CUHK Joint Lab, The Chinese University of Hong Kong,
[2]S-Lab, Nanyang Technological University, [3]Shanghai AI Laboratory, [4]SenseTime Research,
[5]Centre of Perceptual and Interactive Intelligence
{wt020, wt019, dhlin}@ie.cuhk.edu.hk, junzhe001@e.ntu.edu.sg,
{liang.pan, ziwei.liu}@ntu.edu.sg

## Abstract

Chamfer Distance (CD) and Earth Mover's Distance (EMD) are two broadly adopted metrics for measuring the similarity between two point sets. However, CD is usually insensitive to mismatched local density, and EMD is usually dominated by global distribution while overlooks the fidelity of detailed structures. Besides, their unbounded value range induces a heavy influence from the outliers. These defects prevent them from providing a consistent evaluation. To tackle these problems, we propose a new similarity measure named **D**ensity-aware **C**hamfer **D**istance (DCD). It is derived from CD and benefits from several desirable properties: **1)** it can detect disparity of density distributions and is thus a more intensive measure of similarity compared to CD; **2)** it is stricter with detailed structures and significantly more computationally efficient than EMD; **3)** the bounded value range encourages a more stable and reasonable evaluation over the whole test set. We adopt DCD to evaluate the point cloud completion task, where experimental results show that DCD pays attention to both the overall structure and local geometric details and provides a more reliable evaluation even when CD and EMD contradict each other. We can also use DCD as the training loss, which outperforms the same model trained with CD loss on all three metrics. In addition, we propose a novel point discriminator module that estimates the priority for another guided down-sampling step, and it achieves noticeable improvements under DCD together with competitive results for both CD and EMD. We hope our work could pave the way for a more comprehensive and practical point cloud similarity evaluation. Our code will be available at https://github.com/wutong16/Density_aware_Chamfer_Distance.

## 1 Introduction

Point cloud as one of the fundamental 3D representations is attracting increasing attention due to its efficiency, flexibility, and direct connection to real-world objects through 3D scanning devices. It has been employed in a wide range of application scenarios and studied for various tasks [1, 4, 7, 11, 25, 33, 35, 27, 21, 36, 38]. A proper similarity measure between two point clouds is always a crucial aspect for both guiding the training process and providing a fair and reasonable evaluation. However, this is a challenging design considering the unordered and irregular data form and varying point numbers.

Chamfer Distance (CD) and Earth Mover's Distance (EMD) are two of the most universally acknowledged metrics in various point cloud tasks. CD is a nearest-neighbour-based method and benefits from its efficient computation and flexible applicability for point sets with different point numbers. EMD

35th Conference on Neural Information Processing Systems (NeurIPS 2021).

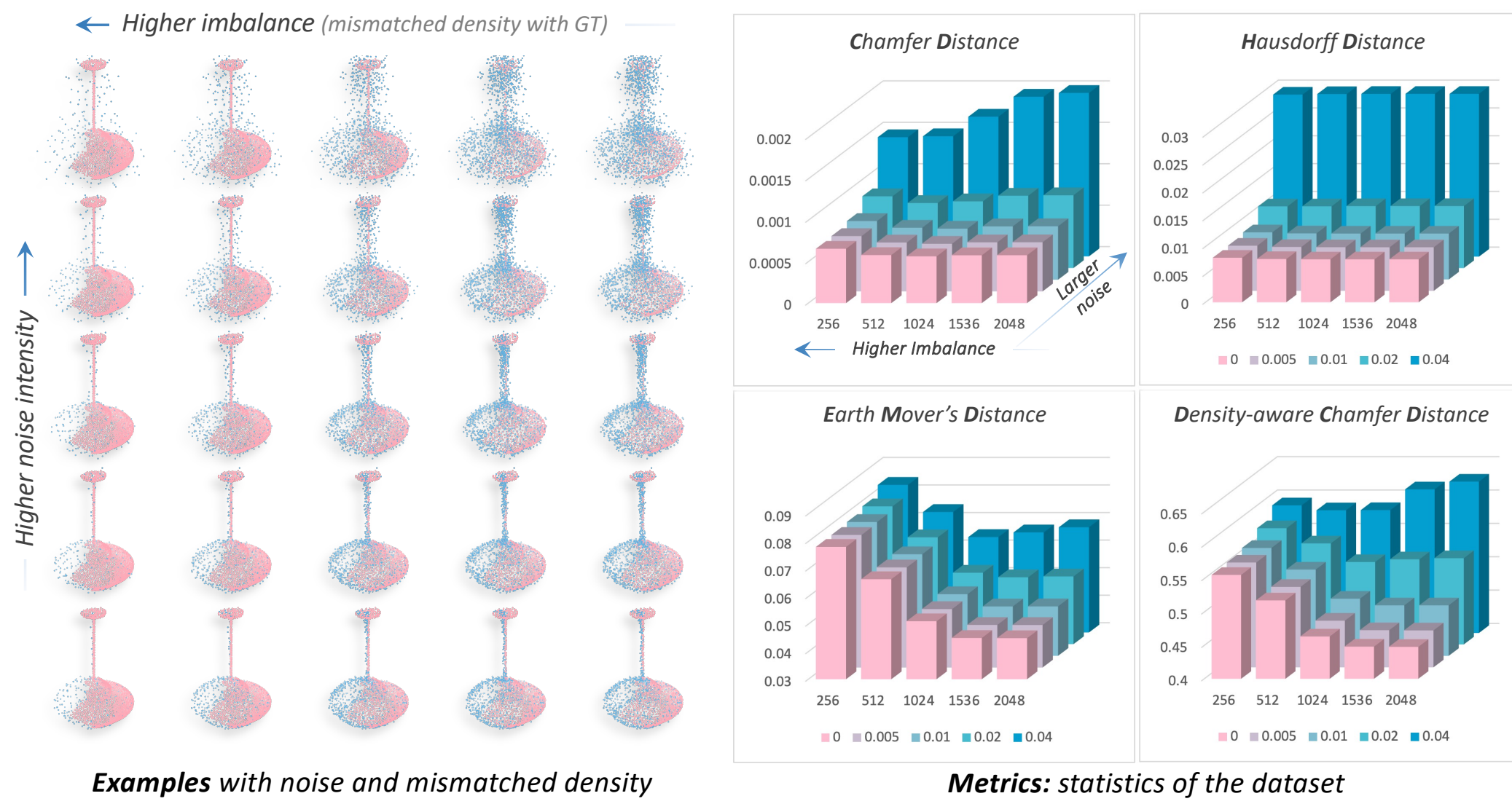

Figure 1: To generate the examples on the left, a complete yet noisy shape (blue) with $n$ points is combined with a partial yet clean shape (pink) with 2048 points and then down-sampled back to 2048 points via FPS. Four point set distances are calculated between the generated shape and the ground truth. The results are averaged over the whole dataset to get the 3D histograms on the right: CD and HD are not sensitive to the mismatched density while highly influenced by noise when the intensity achieves a certain level; EMD and DCD share a similar pattern as $n$ changes, but DCD is more sensitive to the noise which also represents detailed structures.

relies on solving an optimization problem to find the least expensive one-to-one transportation flow between two point sets. Although sometimes considered to be more faithful to visual quality than CD [1, 13], it is significantly more computationally expensive. The desirable properties of a good similarity metric can be different across application scenarios. **1)** For perception and registration, it should be robust to sampling strategies and noisy points on detecting the similarity between the continuous surfaces represented by the discrete points. **2)** For modelling and generation, it should be stricter with the quality of local point distributions, which is also crucial for visual quality. We mainly focus on the second aspect in this paper.

Take a closer look at the two metrics above. The formulation of CD sometimes suffers from its intrinsic deficiencies: being insensitive to different density distributions while significantly influenced by outliers [22]. In particular, we specialize one concept closely related to visual quality, namely *a matched density distribution between two point sets*. This can also be denoted as 'balance', assuming the ground truth is uniformly distributed. We visualize the issue in Fig. 1 through the task of point cloud completion: 1) under a low noise level, CD hardly changes with the imbalance ratio; 2) it increases dramatically once the noise intensity rises to some extent; 3) under a high noise level, higher imbalance even results in a lower CD, which leads to a tricky operation to reduce CD for the task: increasing the point density in the seen area with assured accuracy while reducing it in the unseen area to lower down the risk from abnormal points. But severe imbalance also significantly affects the global appearance. On the contrary, EMD can steadily detect the change of distribution in Fig. 1. However, the requirement for one-to-one mapping is usually over harsh. Consequently, the optima of the transportation problem is dominated by the global distribution while ignoring the fine-structured local details [5], as to be described in Sec. 3. Therefore, both CD and EMD are not ideally suitable for evaluating the quality of the generated shapes.

In this work, we mainly focus on CD's injustice as an evaluation metric and propose a new similarity measure named **D**ensity-aware **C**hamfer **D**istance (DCD) to tackle the challenges above. Specifically, DCD is derived from the original CD, while it benefits from a higher sensitivity to distribution quality through a fraction term of query frequency and a higher tolerance to outliers through an approximation of Taylor Expansion. It shares a similar trend with EMD under a varying point distribution while being more computationally efficient and better at capturing the details. Moreover, due to their different focus, CD and EMD often encounter divergence when evaluating different methods, which makes them less reliable as consistent metrics, as to be demonstrated in Sec. 5. We empirically observe that DCD usually provides a more consistent and reliable evaluation, especially when the

Table 1: Comparison of properties among different metrics.

| Metrics | assignment | efficient | bounded | density-aware | detail-aware |
|---|---|---|---|---|---|
| CD | nearest neighbour | ✓ | × | × | ✓ |
| EMD | optimization | × | × | ✓ | × |
| DCD | nearest neighbour | ✓ | ✓ | ✓ | ✓ |

results of CD and EMD contradict each other. Note that the proposed metric is also beneficial at dealing with ground truth with a non-uniform distribution such as curvature-based sampling, as to be discussed in the supplementary material.

Furthermore, we analyze the property of DCD as a loss function and compare it with L1 and L2 versions of CD. We introduce minor adjustments to it that are critical for a better training process. We then propose to make better use of the information from the query frequency mentioned above and design an MLP-based point discriminator inspired by the recent success of implicit functions [18, 14, 19]. It is further integrated to our balanced design for a two-stage completion framework, where the output of the module can be viewed as the "importance" of each point and serve as priority for a following step of down-sampling. Finally, the guided down-sampling operation benefits from removing outliers and preserving the critical points.

Extensive investigations are provided for the comparison among different metrics and methods for the task of point cloud completion. Experimental results validate that the proposed metric, Density-aware Chamfer Distance, successfully overcomes the aforementioned issues of CD. DCD can provide a more reliable evaluation when CD and EMD contradict each other, and it is proved to be more faithful to visual quality in Sec. 5 and a user study in the supplementary material. We validate its capacity as a loss function on PCN [35] and VRCNet [17], showing that it not only helps reduce DCD itself but also significantly lowers down the EMD metric and surprisingly reduces CD as well compared with the network trained with CD. Our proposed balanced design also gains noticeable improvement under the new metric, competitive results for both CD and EMD, and superior visual quality in the experiments.

## 2 Related Works

**Point Cloud Completion.** Point cloud completion aims to recover a complete shape based on a partial observation. Earlier works represent shapes with voxels [3, 6, 20], while PCN [35] first proposes to use raw point data and leverages an encoder-decoder structure to generate a global-feature-based coarse shape followed by the folding-based up-sampling [34]. Following works enhance the feature representation by techniques like attention mechanisms [28, 17, 16], hierarchical aggregation [8, 37], and grid structure for cubic feature sampling [32], etc.; the decoding process can also be examined to leverage, for example, a tree-structure [23], iterative refinement [26, 31], multiple patches generation [13], or separated prediction for the seen and unseen [37], etc. These works use either CD or EMD for evaluation, yet the two may not be consistently satisfied due to their different concerns, as to be shown in Sec. 5, and thus a more comprehensive metric is essential for a fair and reliable comparison.

**Point Cloud Distance.** The term "distance" refers to a non-negative function that measures the dissimilarity between two point sets. Considering the unordered structure of point clouds, the shape-level distance usually comes from statistics of pair-wise point-level distances based on certain assignment strategy. Chamfer Distance (CD) is one of the most widely used metrics based on the nearest neighbour (Eqn. 1). There are variants of it used for training [34, 4] and some other similarly formed distances like Hausdorff [9, 2, 29]. Another generally adopted metric is Earth Mover's Distance (EMD), which relies on solving an optimization problem to find the best mapping function from one set to the other. It is sometimes considered to be more rational than CD [1, 13], but is much more computationally expensive. Recently, Urbach *et al* [24] propose DPDist, which compares point clouds by measuring the distance between the surfaces that they were sampled on. However, it is estimated by a network rather than a mathematical formulation, making it inconvenient and potentially unstable to be adopted into various tasks. Another closely related work is by Nguyen *et al* [15], who propose the sliced Wasserstein distance that has equivalent properties as EMD and similar computational complexity to CD, with the Monte Carlo method involved for approximation. In comparison, we start from the opposite point of view by deriving a new formulation based on CD

and result in a clean and explicit expression. Similarly, our DCD also shares the properties of EMD in many cases (Fig. 1), while it detects the detail preserving issue better than EMD.

## 3 Density-aware Chamfer Distance for Point Sets

### 3.1 Preliminaries

Chamfer Distance between two point sets $S_1$ and $S_2$ is defined as:

$$d_{CD}(S_1, S_2) = \frac{1}{|S_1|} \sum_{x \in S_1} \min_{y \in S_2} ||x - y||_2 + \frac{1}{|S_2|} \sum_{y \in S_2} \min_{x \in S_1} ||y - x||_2. \tag{1}$$

Each point $x \in S_1$ finds its nearest neighbour in $S_2$ and vice versa; all the point-level pair-wise distances are averaged to produce the shape-level distance. The simple and flexible formulation generalizes well across many tasks. Earth Mover's Distance is defined as:

$$d_{EMD}(S_1, S_2) = \min_{\phi: S_1 \rightarrow S_2} \sum_{x \in S_1} ||x - \phi(x)||_2. \tag{2}$$

It relies on solving an optimization problem that finds a one-to-one bijection mapping $\phi : S_1 \rightarrow S_2$, thus only applicable when $|S_1| = |S_2|$. The pair-wise distances are then calculated between $x$ and $\phi(x)$. As computing the optimal mapping is computationally expensive and even hardly affordable, several approximation schemes [13, 12] have been developed to relieve the computation burden.

### 3.2 Density-aware Chamfer Distance

**Formulation and Interpretation.** As discussed in Sec. 1, CD is not a comprehensive metric for evaluating visual quality for the generation tasks, e.g., point cloud completion. We explain this from its formulation: **1)** the square operation makes it intensively influenced by outliers, and the evaluation results have a huge varying range across the dataset; **2)** the nearest point query operation makes it less sensitive to the issue of mismatched density distribution, and hence resulting in a less discriminative evaluation of visual quality. Therefore, we aim to propose a new metric based on the original formulation that not only preserves the capacity of similarity measure but also highly alleviates the problems above, namely Density-aware Chamfer Distance (DCD).

Firstly, CD grows quadratically with point pair distances, which can be dominated by the worst cases and hence overlooking the others. To address this problem, we introduce the first order approximation of Taylor Expansion $e^z = \sum_{n=0}^{\infty} z^n/n!$, *i.e.* $e^z \approx 1 + z$ where $z = -||x - y||_2$. Thanks to the nearest neighbour assignment, the condition that $z \approx 0$ is usually satisfied and the approximation is reasonable. Thus we have:

$$d_{CD}(S_1, S_2) \approx \frac{1}{|S_1|} \sum_{x \in S_1} \min_{y \in S_2} (1 - e^{-||x-y||_2}) + \frac{1}{|S_2|} \sum_{y \in S_2} \min_{x \in S_1} (1 - e^{-||y-x||_2}). \tag{3}$$

Considering the property of the exponential function $e^z (z < 0)$, each point-level distance is mapped to a value between $[0, 1]$. As a result, the formulation also sets a natural boundary of $[0, 1]$ for the overall shape distance. The approximation would be less accurate as the point-level distance gets away from zero, yet it exactly helps to mitigate the over-sensitivity of the outliers by suppressing the square growth. We add another scale factor $\alpha$ as in Eqn. 4 to adjust the sensitivity. The absolute distance value depends on this factor, and we reveal its relative consistency across different choices of $\alpha$ in the supplementary material, and we fix $\alpha = 1000$ for evaluation in this paper.

Secondly, the ambiguity of CD is partially due to its "blindness" that each point only considers its nearest neighbour in the other set while ignoring the surroundings. We denote the nearest neighbour assignment process by "query" for simplicity and present a simple example here: assume $S_1$ to be a uniform point cloud and $S_2$ an in-homogeneous one; consider two points $y_1, y_2 \in S_2$ with $y_1$ located at a sparse area and $y_2$ in a relatively dense area. The calculation of $d_{CD}(S_1, S_2)$ would likely get $y_1$ frequently queried by points in $S_1$ due to the local sparsity, and the case for $y_2$ can be exactly the opposite. Thus we denote that $y_1$ and $y_2$ are not equally critical in representing the shape. Furthermore, assume a subset $S_1^y \subseteq S_1$ so that each point in $S_1^y$ queries $y$ and that $n_y = |S_1^y|$, points in this set are unaware of each other under the formulation of CD, and the contribution of $y$ to each point in $S_1^y$ is not affected as $n_y$ gets larger, which is unreasonable.

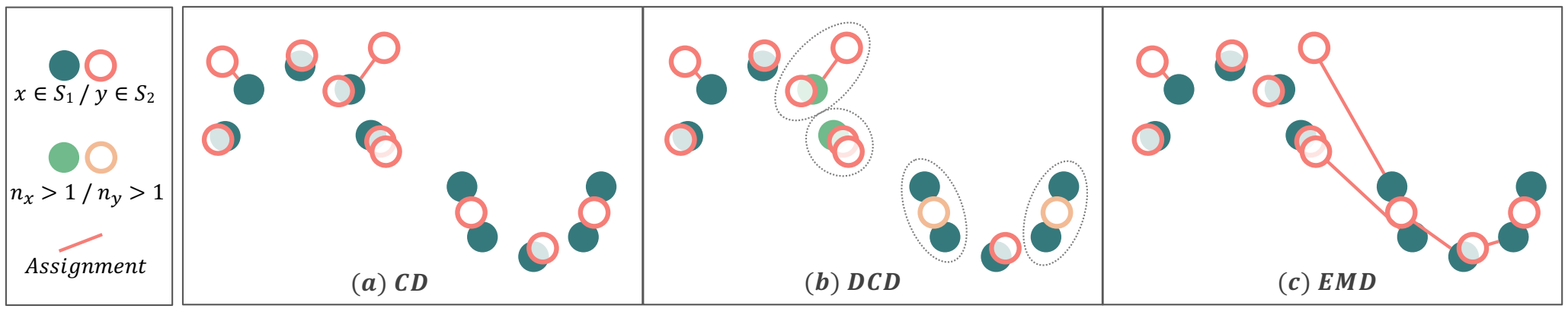

Figure 2: Comparison of assignment strategies and distance calculation. CD and DCD take the nearest neighbour locally, and DCD further considers the point-specific query frequency; EMD forces a one-to-one mapping, and the assigned pair of points may locate far from each other with weaker physical meaning.

Alternatively, we denote that the overall contribution of each point to the evaluation system shall be normalized, and we introduce $1/n_y$ to deal with the case where $y$ is being shared by multiple $x$s:

$$d_{DCD}(S_1, S_2) = \frac{1}{2}\left(\frac{1}{|S_1|}\sum_{x\in S_1}\left(1-\frac{1}{n_{\hat{y}}}e^{-\alpha||x-\hat{y}||_2}\right)+\frac{1}{|S_2|}\sum_{y\in S_2}\left(1-\frac{1}{n_{\hat{x}}}e^{-\alpha||y-\hat{x}||_2}\right)\right), \tag{4}$$

where $\hat{y} = \min_{y\in S_2}||x-y||_2, \hat{x} = \min_{x\in S_1}||y-x||_2$, and $\alpha$ denotes a temperature scalar. Finally, consider the first term without loss of generality, each $y$ contributes $|-\frac{1}{n_y}\sum_{x\in S_1^y} e^{-||x-y||_2}| \in [0,1]$ to the overall distance metric (before averaging). A variants for DCD to deal with point sets with different number of points will be discussed in the supplementary material.

**Comparison among the Distance Metrics.** We conduct a brief comparison of the properties among the three metrics in Table 1, sketching their assignment strategies and computation schemes in Fig. 2. As discussed above, both CD and DCD assign point pairs by querying the nearest neighbours, while DCD further considers the point-specific query frequency $n_y$, and incorporates the property of density distribution into the measurement. In comparison, EMD naturally forces an equal query frequency of 1 for each point via the mapping function, and the metric is highly sensitive to the global point distribution. However, the harsh constraint not only imposes a significant increase in computational cost but also tends to sacrifice its attention to visual quality for optimal mapping. As shown in Fig. 2, the assigned pair of points could be located far from each other, and the distance is less physically meaningful. Experimental results in Sec. 5 would also show that an over compacted and smoothed shape can be favored by EMD despite its loss of detailed structure.

In brief, DCD takes a step from CD and attempts to provide a rationale bridge towards EMD for a better sense of point distribution rather than being blinded by its nearest neighbour. Compared with EMD, it is not only more efficient but also stricter with local structures. A balanced distribution and good preservation of detailed structures are both important factors for the visual quality of the completion result. More examples are to be shown in Sec. 5 for better illustration.

### 3.3 Application as an Objective Function

**Gradient Analysis and Comparisons.** Besides its usage as an evaluation metric, DCD is also expected to serve as an objective function to guide the training process. Considering the same nearest neighbour assignment, our analysis below will mainly focus on comparing CD (denoted by CD-T) and its L1 version (denoted by CD-P), which is also widely adopted for training. The exponential formulation in DCD takes in the L2 distance and modifies the gradient curve while bounding the loss value between [0,1] at the same time.

We visualize the loss value and gradient curves of the three in Fig. 3 (a and b). The gradient by CD-T grows linearly with the distance $l$ of a single point pair: it becomes rather small when $l$ is close to zero and performs a heavy punishment on points with a large $l$. The CD-P loss, on the contrary, produces a constant gradient. For the DCD loss, it would first rise and then approach 1 as $l$ increases, and its gradient can be calculated as: $\delta\hat{d}_{DCD}(l)/\delta l = 2\alpha l e^{-\alpha l^2}/n$, where $\hat{d}_{DCD}(l)$ denotes the contribution of one point pair with a distance of $l$, and $n = n_{\hat{y}}$. The gradient first rises and then falls to zero, indicating that the loss is only effective to point pairs whose distance lies within a certain range. Similar to CD-T, it becomes small when $l$ reduces to zero, which is more reasonable than the

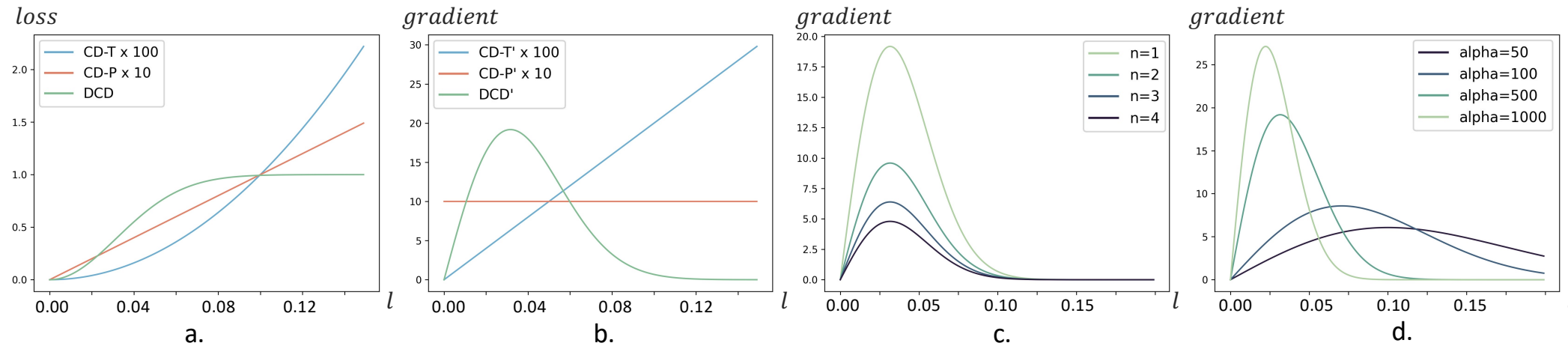

Figure 3: **a** presents the loss curves of CD-T, CD-P, and DCD and **b** presents the gradient curves of them; **c** and **d** visualizes the gradient for DCD with different $n$ and $\alpha$, respectively.

constant gradient by CD-P, while it is also bounded by a maximum and will not be so harsh to the points with an extremely large $l$ and thus stabilizes the training process.

**Adjustment on DCD for Training.** There are two characteristic components that decide the property of the curve, namely the hyper-parameter $\alpha$ and the query frequency $n$. As shown in Fig. 3(c), a larger $\alpha$ promotes a higher peak and a smaller range for non-zero gradient, and we need to set a proper $\alpha$ during training based on the practical distance distribution. We empirically find that setting $\alpha \in [40, 100]$ best promotes the training. A larger $n$ indicates a more serious mismatching of density in a local region and leads to a higher loss value. When viewing both $n$ and $l$ as variables, they are encouraged to be smaller to lower down the loss, but $n$ is not differentiable and cannot be directly changed by backpropagation. Meanwhile, a higher $n$ reduces the gradient to $l$ linearly (Fig. 3(d)), which prevents one point from moving towards an over-dense region, while it may also block the training process when the gradient to $l$ gets too small. As a result, we introduce another hyper-parameter $\lambda \in [0, 1]$ and replace $n_{\hat{x}}$ and $n_{\hat{y}}$ in Eqn. 4 with $n_{\hat{x}}^{\lambda}$ and $n_{\hat{y}}^{\lambda}$ to get the practical loss function for training.

# 4 Incorporating Balanced Design in Point Cloud Completion

## 4.1 Point Cloud Completion Framework

We adopt a typical two-stage coarse-to-fine completion pipeline [35, 13, 26, 17] (Fig. 4(d)). In the first stage, we extract a global feature from the partial observation and generate a complete yet coarse point cloud; in the second stage, we then introduce local features with abundant geometry information and obtain the final output with more details, higher visual quality, and high fidelity to the input. More details of the network architecture and training loss are included in the supplementary material.

We would like to highlight two observations regarding the point distributions here. First, a mean shape usually exists for each category, and there is an obvious imbalance of density for different regions according to how commonly they are shared across the dataset (Fig. 4(a)); second, the evaluation results by CD does not perfectly align with human assessment: the trick that place more points in the seen region with high confidence usually boosts the CD performance while it introduces imbalanced distribution and hurts the benign global distribution at the same time.

To address the problems above, we propose a simple yet effective method based on a current SoTA approach [17] that significantly boosts both the qualitative and quantitative results. Specifically, we tried our new metric as the objective function to replace $L_{CD}$; we then propose a novel point discriminator which is supervised by a carefully designed density-aware signal and estimates the importance of each point; the output from the point discriminator is used as a priority for a final guided down-sampling process that helps remove outliers and maintain a balanced distribution.

## 4.2 Point Discriminator

The generated points in a shape are usually not equally important. We can roughly group them into three categories with the help of query frequency $n$ introduced in Sec. 3: 1) when $n > 0$, the points with a higher $n$ often lie in a sparse area and play a critical role in representing the shape; 2) when $n = 0$, the point can lie on the ground truth surface while in an over-populated region and become relatively unimportant; 3) another case with $n = 0$ is that the point locates far from the underlying surface and can be regarded as an outlier that hurts the overall appearance. There are

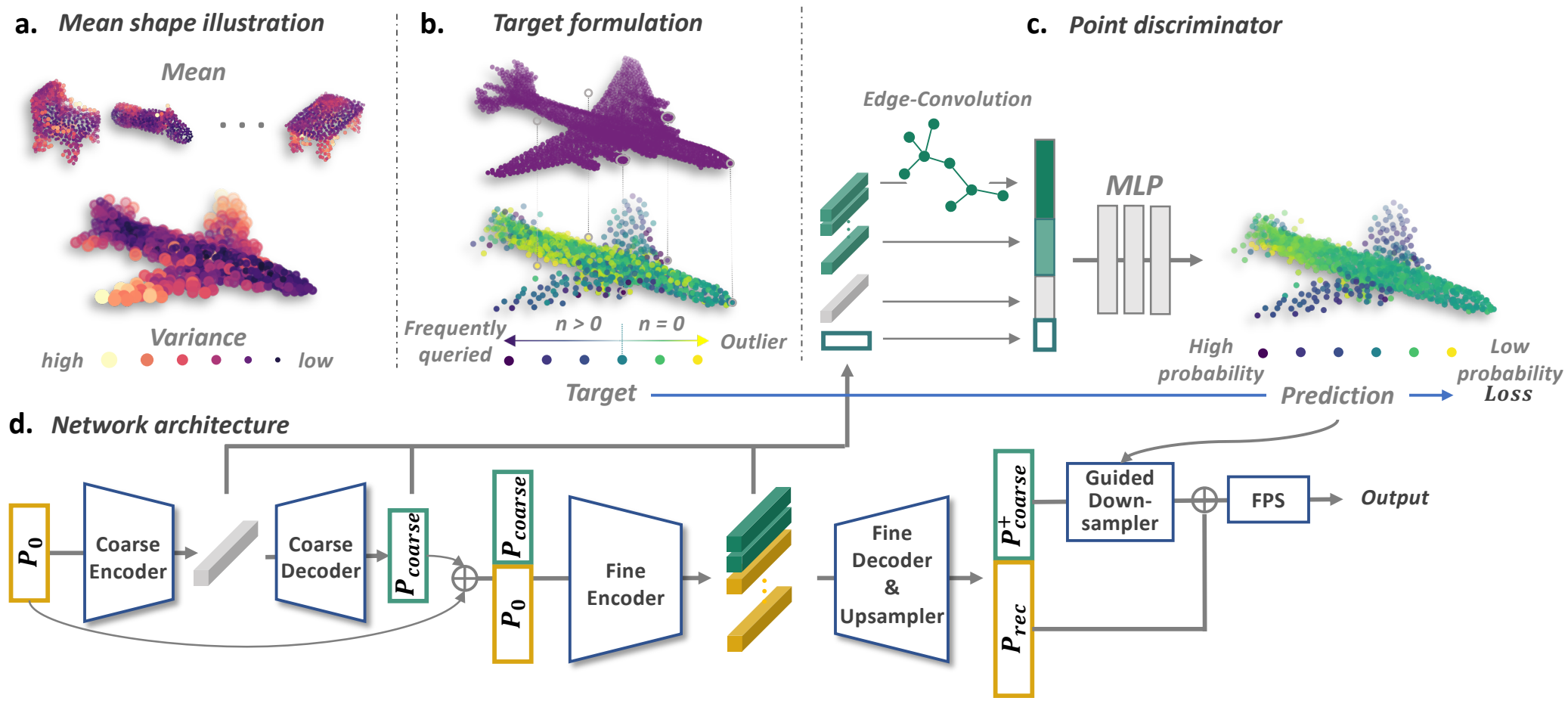

Figure 4: **a.** visualization of mean shape over the dataset; **b.** visualization of $g(x)$ in Eqn. 5 at instance level; **c.** module structure of the point discriminator; **d.** two-stage framework with guided down-sampling.

no hard boundaries among these cases above, but we carefully design a formulation $g(x)$ in Eqn. 5 to map them into a proper value range for learning. Specifically, suppose we train the module with the ground truth $P_{gt}$ and the coarse output $P_{coarse}$, when $n > 0$, we adopt a logarithmic function for $n$ and introduce the expected frequency $|P_{gt}|/|P_{coarse}|$ as the denominator; we add a bias of 1 to ensure $g(0) = 0$ at the boundary and finally invert the sign to distinguish from the following cases. When $n = 0$, we denote that the distance from a point $x$ to $P_{gt}$ together with a scaling factor $t$ can serve as a suitable indicator to distinguish between case 2) and 3). To this end, we formulate a conjoint objective $g(x)$ as below:

$$g(x) = \begin{cases} \min_{y \in P_{gt}} ||x - y||_2 \cdot t & n_x = 0, \\ -\log_2(\frac{|P_{coarse}|}{|P_{gt}|} n_x + 1) & n_x > 0. \end{cases} \tag{5}$$

Encouraged by the recent success of implicit functions [18, 14, 19], we train an MLP module $h$ along with the main network that learns to predict the target function $g(x)$, namely point discriminator. Specifically, it takes the local features of a point and its neighbours for edge convolution; the results are concatenated with the local feature $f^l_x$, global feature $f^g$, and point coordinates $c_x$, denoted by $z = h(f^l_{x'}, f^g, f^l_x, c_x)$, where $x' \in \mathcal{N}_x$ and $\mathcal{N}_x$ denotes the neighbours for point $x$. It outputs a single scalar $z$ which is supervised by the target function $g(x)$ with a regression loss:

$$L_h = \frac{1}{|P_{coarse}|} \sum_{x \in P_{coarse}} ||z - g(x)||_2. \tag{6}$$

The discriminator would be used for deciding point sampling privilege at inference time.

**Guided Down-sampling.** In view of the imbalanced population issue discussed above, an intuitive idea to alleviate it would be to allow a larger number of points generated via up-sampling before reducing to the desired number via Furthest Point Sampling (FPS). However, FPS tends to select more points from outer regions, which increases the risk of including more outliers, and thus this operation usually results in a noisier output. As a result, we aim to take advantage of the point discriminator above for the down-sampling stage at inference time together with FPS. Specifically, for each point $x \in P_{coarse}$, we define a point-wise existing probability by:

$$p(x) = \sigma(-\beta \cdot z - \gamma) = \frac{1}{1 + e^{(\beta \cdot z + \gamma)}}. \tag{7}$$

Let $s$ denote the scale for up-sampling, then $x$ is corresponding to $s$ points in the $P^+_{coarse}$ (up-sampled from $P_{coarse}$) and they all share the same $p(x) \in (0, 1)$, an independent probability of each point being sampled. $P^+_{coarse}$ is first down-sampled in this manner and then combined with $P_{rec}$ (up-sampled from $P_0$) before the final FPS, which ensures that we remain a pre-defined number of points. This process leverages the prior learned during training to effectively reduce the unreasonably located points at inference time (Fig. 7), relieving the side effect brought by FPS operation.

Table 2: Point cloud completion results in terms of CD $\times 10^4$, EMD $\times 10^2$, and DCD, lower is better.

| Methods | Metrics | airplane | cabinet | car | chair | lamp | sofa | tabel | watercraft | bed | bench | bookshelf | bus | guitar | motorbike | pistol | skateboard | Avg. |
|---|---|---|---|---|---|---|---|---|---|---|---|---|---|---|---|---|---|---|
| PCN | CD | 4.50 | 8.83 | 6.41 | 13.01 | 21.33 | 9.90 | 12.86 | 9.46 | 20.00 | 10.26 | 14.63 | 4.94 | 1.73 | 6.17 | 5.84 | 5.76 | 9.78 |
| | EMD | 4.70 | 7.99 | 5.75 | 6.90 | 11.99 | 5.32 | 6.60 | 5.40 | 9.84 | 4.85 | 7.87 | 5.24 | 10.56 | 4.93 | 4.86 | 5.59 | 6.80 |
| | DCD | 0.478 | 0.519 | 0.490 | 0.617 | 0.710 | 0.552 | 0.559 | 0.580 | 0.662 | 0.562 | 0.608 | 0.429 | 0.446 | 0.548 | 0.491 | 0.445 | 0.553 |
| PCN++ | CD | 4.06 | 9.08 | 6.64 | 13.11 | 19.25 | 9.78 | 14.36 | 9.66 | 22.33 | 9.73 | 15.51 | 5.13 | 1.86 | 6.25 | 5.81 | 4.99 | 10.29 |
| | EMD | 3.44 | 3.75 | **3.15** | 4.65 | 8.00 | 3.56 | 4.69 | 4.22 | 6.13 | 3.85 | 4.39 | **2.62** | 2.78 | 3.60 | 3.71 | 3.07 | 4.27 |
| | DCD | 0.428 | 0.464 | 0.451 | 0.574 | 0.661 | 0.504 | 0.517 | 0.540 | 0.617 | 0.524 | 0.563 | 0.389 | 0.369 | 0.527 | 0.447 | 0.393 | 0.508 |
| TopNet | CD | 4.12 | 9.84 | 7.44 | 13.26 | 18.64 | 10.77 | 12.95 | 8.98 | 19.99 | 9.21 | 16.06 | 5.47 | 2.36 | 7.06 | 7.04 | 4.68 | 10.30 |
| | EMD | 4.89 | 6.30 | 4.07 | 7.01 | 10.75 | 6.47 | 7.50 | 4.68 | 8.09 | 6.27 | 6.80 | 3.50 | 4.21 | 4.26 | 6.02 | 3.49 | 6.18 |
| | DCD | 0.536 | 0.558 | 0.548 | 0.650 | 0.711 | 0.598 | 0.599 | 0.600 | 0.678 | 0.588 | 0.622 | 0.492 | 0.487 | 0.572 | 0.542 | 0.496 | 0.598 |
| MSN | CD | 2.73 | 8.92 | 6.50 | 10.75 | 13.37 | 9.26 | 10.17 | 7.70 | 17.27 | 6.64 | 12.10 | 5.21 | 1.37 | 4.59 | 4.62 | 3.38 | 7.99 |
| | EMD | 2.75 | 4.02 | 3.47 | 4.44 | 6.28 | 3.74 | 4.46 | 3.82 | 5.27 | 3.34 | 4.28 | 2.92 | **2.07** | 3.30 | 3.62 | 2.21 | 3.94 |
| | DCD | 0.404 | 0.509 | 0.516 | 0.537 | 0.539 | 0.532 | 0.498 | 0.515 | 0.574 | 0.471 | 0.541 | 0.458 | 0.388 | 0.491 | 0.463 | 0.422 | 0.499 |
| VRC | CD | 2.20 | **7.92** | 5.60 | **7.49** | **8.15** | **7.45** | **7.52** | **5.20** | **11.90** | **4.88** | **7.39** | 4.53 | **1.15** | 3.90 | 3.44 | 3.22 | **6.09** |
| | EMD | 3.03 | 7.57 | 6.14 | 5.49 | 6.15 | 5.80 | 4.65 | 4.97 | 6.58 | 3.45 | 5.28 | 6.59 | 3.08 | 4.45 | 4.56 | 3.20 | 5.27 |
| | DCD | 0.374 | 0.509 | 0.499 | 0.488 | 0.475 | 0.515 | 0.438 | 0.478 | 0.527 | 0.401 | 0.470 | 0.462 | 0.349 | 0.452 | 0.443 | 0.363 | 0.462 |
| VRC-EMD | CD | 2.72 | 9.03 | 6.58 | 9.93 | 11.53 | 9.38 | 9.80 | 6.71 | 17.22 | 6.88 | 10.34 | 5.32 | 1.39 | 4.47 | 4.62 | 4.69 | 7.87 |
| | EMD | 2.50 | **3.65** | 3.23 | **4.15** | 5.31 | **3.61** | 3.93 | 3.58 | **5.17** | 3.19 | 3.97 | 2.69 | 2.08 | 3.06 | 3.48 | **2.29** | **3.62** |
| | DCD | 0.369 | 0.483 | 0.473 | 0.502 | 0.499 | 0.509 | 0.450 | 0.478 | 0.547 | 0.423 | 0.487 | 0.424 | 0.349 | 0.445 | 0.430 | 0.370 | 0.461 |
| Our | CD | **2.22** | 8.00 | **5.41** | 7.88 | 8.28 | 7.94 | 8.89 | 5.46 | 14.76 | 5.78 | 9.37 | **4.44** | 1.30 | **3.59** | **3.43** | **2.39** | 6.51 |
| | EMD | **2.29** | 4.43 | 3.46 | **3.92** | **4.98** | 3.98 | **3.89** | **3.51** | 5.34 | **3.13** | **3.91** | 3.29 | 2.21 | **3.02** | **3.38** | 2.39 | 3.67 |
| | DCD | **0.335** | **0.447** | **0.427** | **0.451** | **0.445** | **0.469** | **0.423** | **0.426** | **0.504** | **0.399** | **0.453** | **0.382** | **0.336** | **0.401** | **0.365** | **0.345** | **0.420** |

# 5 Experiments

**Dataset.** We use the recently proposed MVP Dataset [17] for our study and experiments. It is a multi-view partial point cloud dataset covering 16 categories with 62,400 and 41,600 pairs for training and testing, respectively. It renders the partial 3D shapes from 26 uniformly distributed camera poses for each 3D CAD model selected from ShapeNet [30], and the ground truth point cloud is sampled via Poisson Disk Sampling (PDS).

**Comparison Methods and Metrics.** We include the following methods for comparison in the main experiments: PCN [35], TopNet [23], MSN [13], and VRCNet [17] (with the PSK module discarded to improve efficiency while slightly scarifying performance). PCN++ is a simple extension of PCN [35] that generates the double number of points for training and down-sampled to the required point number at inference time. We report per-class results on CD, EMD, and DCD for a clear comparison across methods and a view of the different properties of these metrics.

**Implementation Details.** All the models are trained using the Adam optimizer [10] with the learning rate initialized at $1e^{-4}$ and decayed by 0.7 every 40 epochs. We use a batch size of 32 and a total epoch of 80. We set $\alpha = 1000$ for the evaluation of DCD, and $\alpha \in [40, 100]$ for training. We set $\lambda \in [0, 0.5]$ and $\beta = 9, \gamma = 1$ for our approach in the main experiments. Our work is implemented with PyTorch and is run on a Tesla V100 GPU.

## 5.1 Comparison of the Methods

The main experimental results for point cloud completion on MVP dataset are reported in Table 2. Note that most networks are trained with the CD loss except when specified. Early works, PCN [35] and TopNet [23] have relatively high loss for all the evaluated metrics, CD, EMD, and DCD. PCN with an additional up-sampling step, namely PCN++, notably lowers down EMD and DCD, while marginally raising CD for sacrifice; MSN [13] benefits from the multi-surface design and obtains low EMD and DCD results, while its CD performance is not that satisfying. The previous SoTA method, VRCNet [17] outperforms the other methods under the metric of CD and DCD, but we observe that its EMD loss is surprisingly high. We further replace the CD loss with EMD when training VRCNet (denoted as VRC-EMD), and it achieves the lowest EMD yet obviously higher CD than the original version. In comparison, our method reports the lowest DCD, second-lowest CD, and comparable EMD with VRC-EMD. Qualitative results (Fig. 5) show that our results are apparently superior in both the global balanced point distributions and local structures. A user study in the supplementary material will further validate that our method t benefits from a higher visual quality.

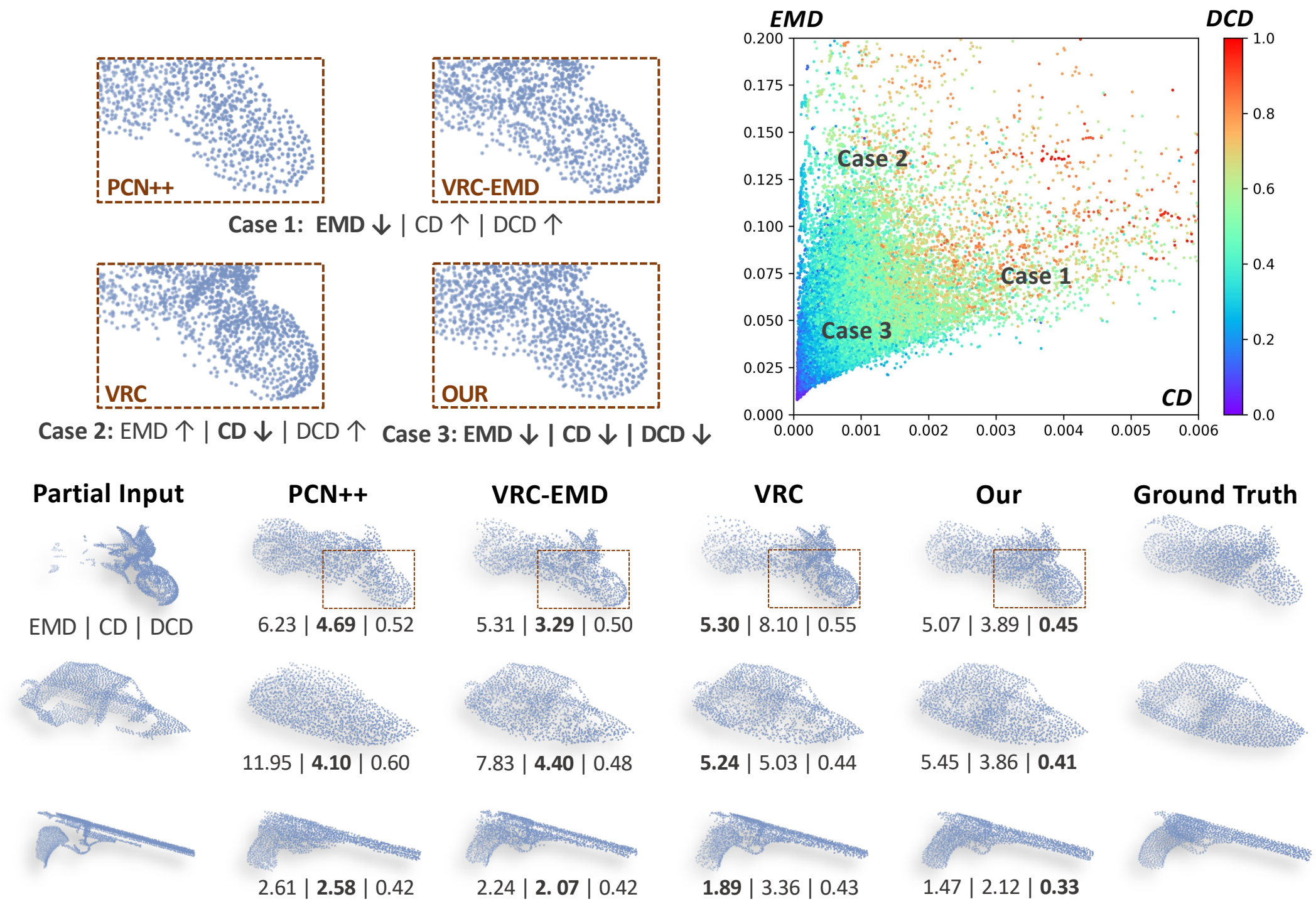

Figure 5: Comparison of CD, EMD, and DCD through examples and statistics. **Below:** examples from different methods that represent three typical cases: 1) EMD is low while CD is high; 2) CD is low while EMD is high; 3) both CD and EMD are low. DCD can only be lowered down in the third case. **Upper left:** A close shot at the cases above. **Upper right:** visualization of the three metrics on the test set, we can observe that 1) the positive correlation between CD and EMD is weak and the scatter points form a fan-shaped area, and 2) DCD (denoted by the point color) becomes lower towards the original point where both CD and EMD are low.

### 5.2 Comparison of the Metrics

**Consistency and Reliability.** When taking a closer look at all the CD-EMD-DCD tuples (either for each category or for the averaged results) in Table 2, the results by different metrics do not exhibit a clear positive correlation. But we also observe that for those that have a similar value of CD, DCD is usually dominated by their EMD performance, and for those with similar EMD results, DCD is highly correlated with CD. Although the law is not strictly held or theoretically proved, the scatter plot by each instance from different methods in Fig. 5 (upper right) also supports our empirical observation. It reveals that since CD and EMD focus on different aspects of the point cloud, inconsistency and confusion may exist for the similarity measurement, which prevents either of them from being a comprehensive metric. In comparison, DCD reflects the behaviors of them both and could only be reduced when both CD and EMD are relatively low so that it serves as a more consistent, stable, and comprehensive metric, especially when CD and EMD encounter disparity. Our user study in the supplementary material further indicates that DCD is a more faithful metric to visual quality.

**Bounded Distance.** One advantage of DCD is its bounded value range, which promotes an equal consideration for all samples in the dataset at the evaluation stage. In contrast, CD and EMD are dominated by the worst cases in the dataset due to their unbounded nature, as shown in Fig. 6. Specifically, for CD, 80% of the normalized loss accumulation is contributed by only top 50% of the samples ranked by the shape-level distance, and 50% contributed by the top 25%; a similar case is also observed for EMD. This property makes them fail to comprehensively evaluate the quality of all samples in the dataset. Our DCD overcomes this problem by assuring a clear [0,1] boundary for the distance value, enabling more comprehensive and stable statistical results over the whole test set.

### 5.3 Ablation Study

We study the effectiveness of each component in our method separately, including DCD loss, the additional point up-sampling (PU), and the guided point down-sampling (GPD). As shown in Table 3,

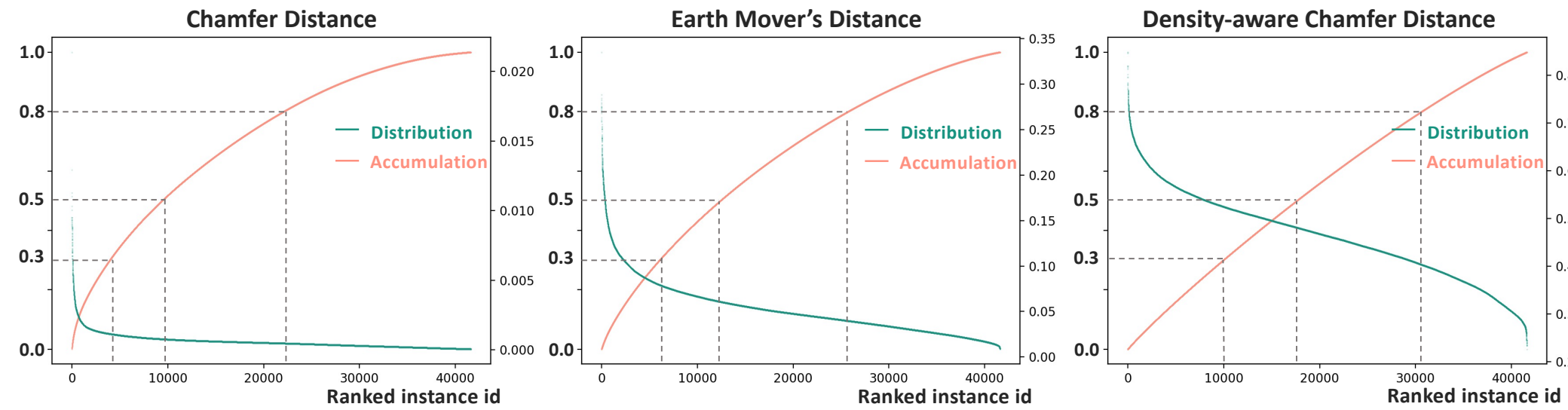

Figure 6: Distribution and accumulation (normalized to 1) of evaluation results per-shape in the test set.

Table 3: Ablation Study.

| DCD | PU | GPD | CD | EMD | DCD |
|---|---|---|---|---|---|
| | | | Metric | | |
| | | | 6.09 | 5.27 | 0.462 |
| ✓ | | | **5.85** | 5.14 | 0.457 |
| | ✓ | | 6.91 | 3.78 | 0.425 |
| ✓ | ✓ | | 6.65 | 3.68 | 0.422 |
| ✓ | ✓ | ✓ | 6.51 | **3.67** | **0.420** |

Table 4: Evaluation results on three metrics when trained with each of them. * denotes applying $L_{EMD}$ on the final outputs while $L_{CD}$ on intermediate ones.

| Model | PCN [35] | | | VRC [17] | | |
|---|---|---|---|---|---|---|
| Metric | $L_{CD}$ | $L^*_{EMD}$ | $L_{DCD}$ | $L_{CD}$ | $L^*_{EMD}$ | $L_{DCD}$ |
| $CD$ | 9.78 | 10.70 | **9.36** | 6.09 | 7.87 | **5.85** |
| $EMD$ | 6.80 | **3.97** | 4.71 | 5.27 | **3.62** | 5.14 |
| $DCD$ | 0.553 | 0.537 | **0.526** | 0.462 | 0.461 | **0.457** |

leveraging $L_{DCD}$ loss usually outperforms the same model trained with $L_{CD}$ in all the three metrics; the additional point up-sampling (PU) significantly reduces EMD while it increases CD at the same time; after guided down-sampling (GPD) is applied to the model with both $L_{DCD}$ and PU, we achieve the lowest EMD and DCD and a relatively low CD.

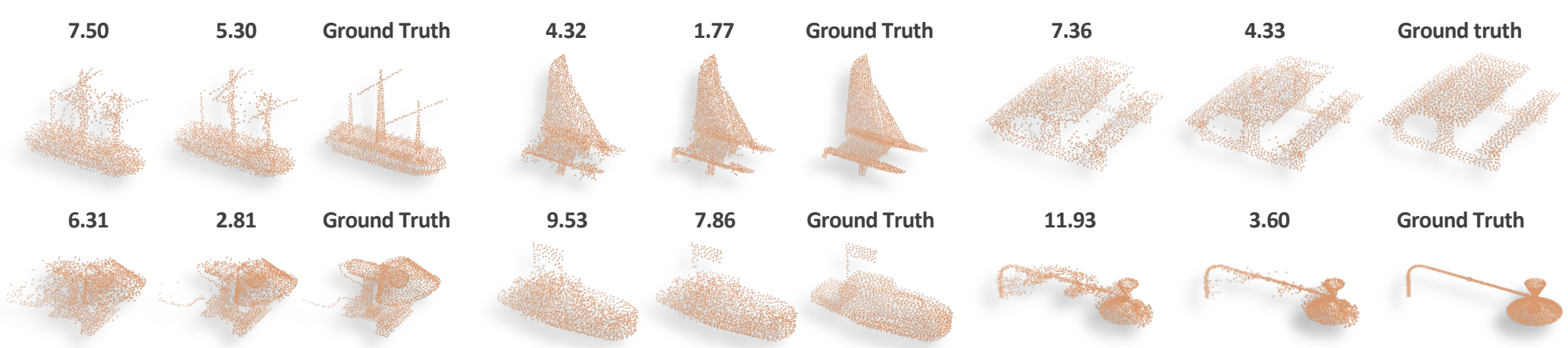

Figure 7: Examples of guided down sampling (from left to right) and evaluation in terms of CD $\times 10^4$.

### 5.4 Performance as the Loss Function

We evaluate the effectiveness of DCD as a loss function by training networks with each of the three metrics (denoted by $L_{CD}$, $L_{EMD}$, and $L_{DCD}$) and evaluate on all of them (denoted by CD, EMD, and DCD). We conduct experiments on the baseline model PCN [35] and the SoTA VRCNet [17], as shown in Table 4. Compared with $L_{CD}$-trained networks, training with $L_{EMD}$ produces the lowest EMD and a lower $d_{DCD}$, yet it suffers from non-negligible scarification of $CD$ and much heavier computational burden, which is to be discussed in the supplementary material; training with $L_{DCD}$ will produce the lowest $DCD$, significantly reduce $EMD$, and can even slightly reduce $CD$ than the $L_{CD}$-trained models (especially on PCN). The time consumption is also comparable with $L_{CD}$, which further validates DCD's convenience and superiority as an objective function. We set $\alpha = 50$ or 100 here with $\beta = 0$ and replace all the $L_{CD}$ with $L_{DCD}$. Experiments on how $\alpha$ and $\beta$ affect the performance are included in the supplementary material.

## 6 Conclusion

In this work, we propose a new similarity measure for point clouds named Density-aware Chamfer Distance (DCD). It is bounded in value, effective in computation, and faithful to visual quality by considering both the density distribution and detailed structures. Our method achieves noticeable improvements under DCD and superior visual quality compared with previous works.

**Acknowledgements.** This study is supported in part by the NTU NAP, and under the RIE2020 Industry Alignment Fund – Industry Collaboration Projects (IAF-ICP) Funding Initiative, as well as cash and in-kind contribution from the industry partner(s). It is also supported in part by Centre for Perceptual and Interactive Intelligence Limited, in part by the GRF through the Research Grants Council of Hong Kong under Grants (Nos. 14208417, 14207319 and 14203518) and ITS/431/18FX, in part by CUHK Strategic Fund and CUHK Agreement TS1712093, in part by the Shanghai Committee of Science and Technology, China (Grant No. 20DZ1100800).

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
