# Supplementary Material:
# Density-aware Chamfer Distance as a Comprehensive Metric for Point Cloud Completion

**Tong Wu**[1], **Liang Pan**[2], **Junzhe Zhang**[2,4], **Tai Wang**[1,3], **Ziwei Liu**[2], **Dahua Lin**[1,3,5]
[1]SenseTime-CUHK Joint Lab, The Chinese University of Hong Kong,
[2]S-Lab, Nanyang Technological University, [3]Shanghai AI Laboratory, [4]SenseTime Research,
[5]Centre of Perceptual and Interactive Intelligence
{wt020, wt019, dhlin}@ie.cuhk.edu.hk, junzhe001@e.ntu.edu.sg,
{liang.pan, ziwei.liu}@ntu.edu.sg

## A   Further Analysis of Balanced Chamfer Distance

### A.1   The consistency and choice of hyper-parameter $\alpha$

We introduce a temperature scalar $\alpha$ in Eqn. (4) so that $e^{-\alpha \cdot ||x-y||^2}$ can have a relatively wider varying range. We set $\alpha = 1000$ in the paper and would briefly show why it is a proper value here. Notice that $||x - y||^2$ for each nearest point pair is usually about $10^{-4}$ or $10^{-3}$, thus setting $\alpha = 1000$ maps it to around $10^{-1}$ or $10^0$ where the exponential term has a large gradient. As visualized in Fig. S1(a), either setting $\alpha$ too large or too small would not result in an ideally shaped function. We also regenerate the DCD value matrix under the same settings as in Fig. **??** with different $\alpha$ values, and we calculate the mean and variance accordingly, as shown in Fig. S1(b). A larger $\alpha$ results in a higher mean which is reasonable according to Eqn. (4), and we observe the variance is at its largest with $\alpha = 1000$, which also aligns well with the theoretical analysis.

Furthermore, we track the PCN training loss calculated by DCD with different $\alpha$, as shown in Fig. S1(c), and we visualize the per-instance evaluation results with different $\alpha$ for a well-trained model in Fig. S1(d). The statistical results show that the relative value and trend of DCD is relatively consistent with different data distributions when $alpha$ changes, while their absolute values are different. At evaluation time, we can use the same $\alpha$ (e.g., 1000) for all the methods for a fair comparison.

### A.2   Dealing with mismatched point numbers.

We consider the case where the two point sets $S_1$ and $S_2$ do not have the same number of points, suppose $|S_1| = \eta \cdot |S_2|, \eta > 1$. A naive extension of DCD (Eqn. (4) in Sec. 3 is presented as Eqn. 1, where we add $\eta$ or $1/\eta$ to indicate the one-to-many mapping in this case:

$$d_{DCD}(S_1, S_2) = \frac{1}{2|S_1|} \sum_{x \in S_1} \left( 1 - \frac{\eta}{n_{\hat{y}}} e^{-\alpha||x-\hat{y}||_2} \right) \\ + \frac{1}{2|S_2|} \sum_{y \in S_2} \left( 1 - \frac{1}{\eta \cdot n_{\hat{x}}} e^{-\alpha||y-\hat{x}||_2} \right), \quad (1)$$

The formulation above usually works well in practice, but it may also lead to negative results in the first term when $n_{\hat{y}} < \eta$ and $\frac{\eta}{n_{\hat{y}}} e^{-\alpha||x-\hat{y}||_2} > 1$. We thus propose another variant of DCD in Eqn. 2. Considering **the first term**, since $|S_1| > |S_2|$ and each $y \in S_2$ should naturally be assigned to more than one $x \in S_1$, the decaying term should not follow the tendency of $1/n_{\hat{y}}$, but rather updated to

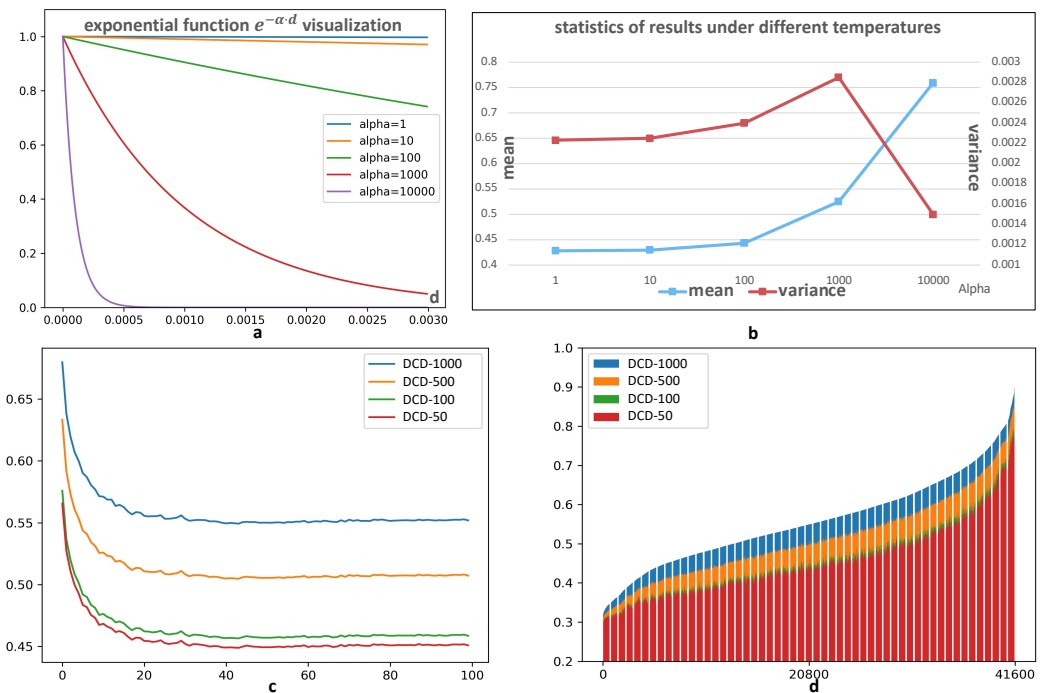

Figure S1: **a** visualization of the exponential function $e^{-\alpha \cdot d}$ with a varying $\alpha$; by empirically setting the value range of $d$ to be similar with $||x-y||^2$ for paired point-wise distance, we found that $\alpha = 1000$ gains a desirable value range by the function. **b** visualization of the statistical results of DCD under different temperatures; a larger $\alpha$ results in a larger mean value and $\alpha = 1000$ gets the highest variance. **c** we track the loss with different $\alpha$ when training a PCN network. **d** the per-instance evaluation with different $\alpha$ for a well-trained network.

$\max(\eta/n_{\hat{y}}, 1)$. On the one hand, the contribution of $\hat{y}$ would not be reduced before the querying frequency of it reaches $\eta$; on the other hand, it should not exceed 1 either, which is important for keeping a non-negative result. As for **the second term**, each $x \in S_1$ is corresponding for more than one point in $S_2$. We take $\overline{\eta} = ceiling(\eta)$ and find $\overline{\eta}$-nearest neighbours for $x$, denoted by $N(y)_{\overline{\eta}}$. And the overall formulation of the variant is:

$$
\begin{aligned}
d_{DCD-E}(S_1, S_2) = & \frac{1}{2|S_1|} \sum_{x \in S_1} \left( 1 - \frac{1}{\max(\eta/n_{\hat{y}}, 1)} e^{-\alpha||x-\hat{y}||_2} \right) \\
& + \frac{1}{2|S_2|} \sum_{y \in S_2} \left( 1 - \frac{1}{\overline{\eta} \cdot n_{\hat{x}}} \sum_{\hat{x} \in N(y)_{\overline{\eta}}} e^{-\alpha||y-\hat{x}||_2} \right).
\end{aligned}
\tag{2}
$$

We use Eqn. 1 in training for the loss between the coarse shape with 1024 points and the ground truth with 2048 points for simplicity.

### A.3 Time Complexity.

EMD relies on solving the linear assignment problem in an iterative approximation manner with a practical time complexity between $O(n^2)$ [4] and $O(n^3)$ [7] and usually $O(n^2)$ memory footprints. We adopt an $O(n)$ memory-efficient implementation by [5] and with the error rate $\epsilon = 0.004$ and an iteration of 3000. The most computationally expensive part for CD and DCD is the nearest neighbour selection, which is usually $O(n)$ for time complexity and can be accelerated by special data structures like KD-tree. We observe that both CD and DCD are significantly efficient to be computed compared with EMD; the running time of EMD is also affected by the distributions of the two point sets, which decides the difficulty of finding the optimal assignment. As shown in Table R1, with the same setting as in Fig. **??**, a higher mismatched distribution results in obviously heavier time consumption, and the noise intensity also influences the results.

Table R1: Time consumption evaluation.

|      | 256   | 512   | 1024  | 1536  | 2048  |
|------|-------|-------|-------|-------|-------|
| CD   | 0.006 | 0.007 | 0.008 | 0.010 | 0.012 |
| EMD  | 0.362 | 0.327 | 0.267 | 0.241 | 0.239 |
| DCD  | 0.013 | 0.013 | 0.013 | 0.013 | 0.013 |

|      | 0     | 0.005 | 0.01  | 0.02  | 0.04  |
|------|-------|-------|-------|-------|-------|
| CD   | 0.008 | 0.008 | 0.008 | 0.008 | 0.008 |
| EMD  | 0.272 | 0.271 | 0.269 | 0.264 | 0.256 |
| DCD  | 0.013 | 0.013 | 0.013 | 0.013 | 0.013 |

## A.4  Evaluation on other tasks and ground truth distributions.

Apart from the task of completion, DCD is also a suitable evaluation metric for tasks like point cloud upsampling or denoising, where the ground truth with desirable point distribution is provided, and the model is expected to generate high-quality point cloud outputs. Taking upsampling as an example, a desirable dense output is expected to be uniform, clean, and faithfully located on the underlying surface, and thus metrics like NUC [11] and uniform loss [3] were proposed to evaluate the distribution uniformity besides Chamfer Distance. However, these metrics usually make strong assumptions that points in a small patch lie on a surface or that there should be an expected number of points in a ball anywhere with a certain radius. And the metrics are sometimes sensitive with the choice of hyper-parameters and the geometry itself. Moreover, they always encourage uniformity rather than the specific distribution of the ground truth, which also limits the application scope.

On the contrary, the density-aware DCD would focus on the faithfulness of the output to the ground truth distribution without any strong assumptions; it is not sensitive to the choice of hyper-parameters; it is beneficial at reflecting the mismatched density in local areas in scenarios where the ground truth is not uniformly distributed for specific purposes (*e.g.*, curvature-based sampling), as shown in Fig. S2. We also perform quantitative evaluation by applying a mixture of curvature-based sampling (for a ratio of $R_c$) and the standard Poisson-Disk Sampling (PDS), while the output is noisy and basically uniform. When $R_c$ changes, DCD and EMD can reflect the increasingly mismatched density, while CD and the uniform loss are not sensitive to it. (Table R2).

Table R2: Evaluation on ground truth with non-uniform sampling. We apply a mixture of curvature-based sampling (for a ratio of $R_c$) and the standard PDS, while the output is noisy and basically uniform. $CD(gt)$ denotes the averaged L2 distance from the ground truth points to their nearest neighbor, and the definition can be extended to $CD(x)$, $DCD(gt)$, and $DCD(x)$. When $R_c$ changes, DCD and EMD can reflect the increasingly mismatched density, while CD and uniform loss are not sensitive to it.

| $R_c$ | $CD(gt)$ | $CD(x)$ | $DCD(gt)$ | $DCD(x)$ | $EMD$ | $L_{uni}(x)$ |
|-------|----------|---------|-----------|----------|-------|--------------|
| 0%    | 1.62     | 2.06    | 3.97      | 3.54     | 2.13  | 19.96        |
| 25%   | 1.68     | 2.21    | 4.24      | 4.00     | 5.64  | 19.96        |
| 50%   | 1.68     | 2.41    | 4.62      | 4.34     | 7.25  | 19.96        |

## A.5  Effect of hyper-parameters in $L_{DCD}$.

We introduce two hyper-parameters in Sec. **??** that would affect the performance of DCD as a loss function, and we conduct experiments accordingly to explore the pattern. As shown in Table R3, a larger $\alpha$ usually promotes a lower DCD, while the performance of CD worsens when $\alpha$ is as large as 1000. $\lambda = 0$ usually results in the best CD results yet with sub-optimal EMD and DCD, while $\lambda = 1$ would obviously hurt CD; $0 < \lambda < 1$ exhibits a trade-off among the metrics.

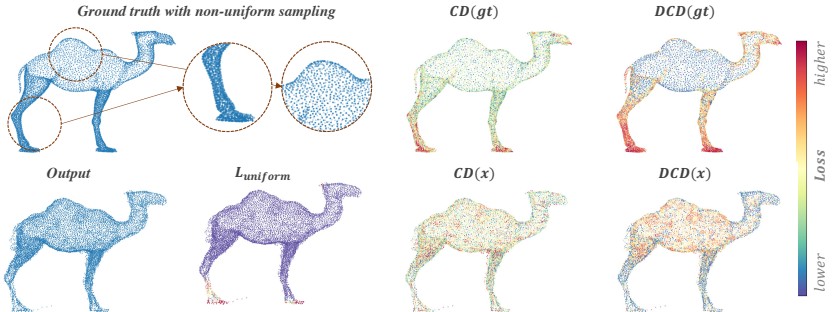

Figure S2: Visualization of the distance contributed by each point. DCD is better at reflecting the mismatched density in local areas between the two sets.

Table R3: The effect of $\alpha$ and $\lambda$ when applying $L_{DCD}$ on PCN [12].

| $\alpha$ | $\lambda$ | CD | EMD | DCD |
|---|---|---|---|---|
| 50 | 0.0 | 9.56 | 4.92 | 0.533 |
| | 0.5 | 9.86 | 4.68 | 0.529 |
| | 1.0 | 10.21 | 4.63 | 0.527 |
| 100 | 0.0 | 9.33 | 4.92 | 0.535 |
| | 0.5 | 9.85 | 4.68 | 0.526 |
| | 1.0 | 9.93 | 4.69 | 0.525 |
| 200 | 0.0 | 9.36 | 4.71 | 0.526 |
| | 0.5 | 9.82 | 4.59 | 0.520 |
| | 1.0 | 10.16 | 4.64 | 0.521 |
| 1000 | 0.0 | 10.14 | 4.72 | 0.516 |
| | 0.5 | 10.56 | 4.72 | 0.516 |
| | 1.0 | 11.12 | 4.96 | 0.519 |

# B  User Study on Visual Quality

In Sec. 5.1 and 5.2, we conduct comparisons on 1) the completion performance among different methods and 2) the characteristics of three metrics. We show that our method gains the best performance under the metric of DCD and that DCD is a more comprehensive measure through several examples. In this section, we further validate the conclusions above via user study. Specifically, we randomly select 25 partial inputs and select five methods (*i.e.*, PCN++ [12], MSN [5], VRC [6], VRC-EMD [6], and our method) to generate 125 completion outputs in total. We exclude the original PCN [12] and TopNet [9] since their performance is not desirable on any of the three metrics; each of the five methods above is able to achieve favorable results under at least one metric, which provides a good playground to evaluate the quality of different metrics.

We invite 15 volunteers to complete the study. For each shape with five generated point clouds in a random order, the volunteers are asked to select a single or multiple (for at most two) items with the highest comprehensive score according to the following evaluation indicators: **1)** The similarity between each option and the ground truth, including the similarity of global shape and the fidelity of local details. **2)** The quality of the point cloud distribution, considering whether there exists a significant shift in the center of gravity and whether there exists obvious clustering or sparseness. Once we get the statistical results, we analyze the data from two points of view as follows.

**Comparison of the methods.** For the output of each method on each shape, we calculate the average ratio of it being selected as the favorite option by the volunteers (e.g., if three people out of fifteen select one method as the best for one shape, the ratio is 0.20). And then, the results are averaged for all the shapes, as shown in Table. R4. Our method outperforms the others by a large margin, indicating that it benefits from high visual quality.

**Comparison of the metrics.** We also leverage the data to evaluate how faithful each metric is to human vision. Specifically, according to each of the metrics, we can get the top-1 completion result

Table R4: The average ratio of each method being considered to have produced the best results.

| Methods | PCN++ [12] | MSN [5] | VRC [6] | VRC-EMD [6] | Ours |
|---|---|---|---|---|---|
| Average Ratio | 0.280 | 0.123 | 0.286 | 0.151 | **0.460** |

Table R5: The faithfulness of best-result-selection between each metric and human users.

| Metrics | CD | EMD | DCD |
|---|---|---|---|
| Average Ratio | 0.574 | 0.437 | **0.623** |

for each shape; similarly, we collect the ratio of it being selected by volunteers as the best option and average the ratio over all the shapes. As shown in Table. R5, DCD gains the highest alignment with human vision.

## C  Network architecture and training details.

### C.1  Review of a Typical Two-Stage Pipeline.

The two-stage coarse-to-fine completion pipeline was first proposed by PCN [12] and improved by a series of its following works [5, 10, 6]. The first stage takes a partial point cloud as input and extracts a global feature $f^g$ by an MLP, *e.g.*, PointNet [8], followed by a decoder to generate a coarse point cloud $P_{coarse}$. Though it maintains a relatively reasonable global shape, $P_{coarse}$ usually fails to capture and depict the details. Therefore, the second stage follows up, which aims **1)** to precisely reconstruct the input point cloud without loss of details, denoted by $P_{rec}$, **2)** to improve the quality of $P_{coarse}$ especially in the unseen part via coordinates adjusting, up-sampling, and probably detail transferring, denoted by $P_{coarse}^+$, and finally **3)** to obtain the final output with desirable visual quality and high fidelity to the original input. Point-wise local features $f^l$ with abundant geometry information encoded are usually involved in this stage. We leverage the same network architecture as VRCNet [6] and adopt the official implementation from their public code [1].

### C.2  Interpretation of the mean shape.

Although CD does not involve any hard assignment between prediction and ground truth point sets, it is observed that the output coordinates from each node of the last Fully Connected (FC) layer have a relatively convergent local distribution. By assuming a Gaussian-like distribution, we visualize their **mean** and **variance** for each category separately across the test set in Fig. **??**, where the mean coordinates form the category-specific **mean shapes**, and the color of each node denotes the node variance [2]. This observation indicates that for the coarse shape, there exists an obvious imbalance of density across different regions according to how commonly they are shared across the dataset. This pattern occurs not only for the *statistical results* but also for *each shape instance* [1, 2].

Another thing to emphasize is the trade-off between accuracy and distribution balance. The first few points located in an unseen area significantly reduce the CD loss, while the marginal gain soon decays with more points predicted there. Considering the side effect that inaccurate points would bring in extra corruption to the overall CD loss, lying more points in the seen region with high confidence usually boosts the CD performance. This trick benefits the CD metric, while it violates the overall distribution and hurts the visual quality at the same time.

### C.3  Training Details

We introduce the regression loss $L_h$ for training the point discriminator in Eqn. (6) (Sec. 4.2), and we would clarify the full training loss to train the two-stage framework here. Specifically, the loss function includes another two parts despite $L_h$: $L_d$ involves multiple paired point cloud distances,

---

[1]https://github.com/paul007pl/VRCNet
[2]The category information is not provided during training, while only used for statistics.

and $L_{KL}$ indicates the KL divergence for the dual-path VAE architecture following [6]. These two terms are formulated as follows:

$$L_d = \lambda_1 \cdot d_{CD}(P_{coarse}, P_{gt}) + \lambda_2 \cdot d_{CD}(P_{coarse}^+, P_{gt}) + \lambda_3 \cdot d_{CD}(P_{fine}, P_{gt}), \quad (3)$$

where $P_{fine}$ denotes the final output after sampling, and $\lambda_1$, $\lambda_2$, and $\lambda_3$ denote the loss weights. Note that we still use the Chamfer Distance as training loss in this paper, and the reason why would be partially explained in Sec. D. We thus have:

$$L_{KL} = -\lambda_{KL} \cdot (\mathbf{KL}(q_\phi(f^g|P_{gt}), \mathcal{N}(0, I)) + \mathbf{KL}(q_\psi(f^g|P_0), q_\phi(f^g|P_{gt}))), \quad (4)$$

where $q$ denotes the encoder for latent distributions with network weights denoted by $\phi$ and $\psi$, $\mathbf{KL}$ denotes the calculation of KL divergence with a loss weight of $\lambda_{KL}$. Finally, the overall loss function is formulated as:

$$L = L_h + L_d + L_{KL}. \quad (5)$$

We set $\lambda_1 = 10$, $\lambda_2 = 0.5$, $\lambda_3 = 1$ and $\lambda_{KL} = 20$ in the experiments.

When we adopt $L_{DCD}$ during network training, we can simply replace all the occurrence of $L_{CD}$ with $L_{DCD}$, while there are also tricks for better performance: 1) we can use different $\alpha$ for different terms in Eqn. 3, e.g., $\alpha = 50$ for the first term and $\alpha = 100$ for the others, which empirically works slightly better than using $\alpha = 50$ or $\alpha = 100$ for all the terms; 2) we can add another L1-version CD (CD-P) along with $L_{DCD}$ for training, which is our implementation to gain the results reported in Table **??**.

## D   Limitations and Future Work

This approach still has some limitations that can be further explored in the future: we investigate the properties of the proposed metric on the task of point cloud completion, while it is actually applicable in many other tasks and scenarios as both evaluation metric and training loss. In the future, we will conduct more experiments to validate the generalization ability of DCD across different tasks.

## E   Broader Impact

A comprehensive, reliable, and effective similarity measure is critical to point cloud analysis. It not only provides a fair comparison among different methods but also encourages the design of algorithms to take more critical factors into consideration, such as preserving accurate local details, keeping a uniform global distribution, and avoiding outliers. As shown in the paper, the broadly used Chamfer Distance and Earth Mover's Distance usually encounter obvious disparity due to their different focus, making it hard to provide a consistent evaluation. It reveals the necessity and importance of formulating a more comprehensive metric to close the gap. We hope that the Balanced Chamfer Distance we propose in this paper can better serve the demands above than the existing metrics, so that hopefully it can encourage a more reasonable evaluation and influence method designs for tasks in point cloud analysis.