# OpenReview forum: "Balanced Chamfer Distance as a Comprehensive Metric for Point Cloud Completion"
_NeurIPS.cc/2021/Conference — NeurIPS 2021 Poster_

### Official Review · Reviewer_ziDa · 2021-07-14

**Rating:** 5
**Confidence:** 4

**Summary:**

The paper proposes a new distance metric for measuring the difference between two point clouds (say P1 and P2). In the past, either Chamfer or Earth-Mover's distance was used. The new distance builds on top of Chamfer, and essentially adds more weight to a point in P1 where multiple points from P2 are matched to that point. The authors claim that the new metric is:

- More distribution sensitive compared to Chamfer.
- More detail sensitive compared to Earth-Mover.
- Efficient to compute compared to Earth-Mover.
- More robust in general.

I must admit that I am not an expert in point cloud matching. But I mostly agree with the authors' assessments.

**Ethics Review Area:**

["I don’t know"]

**Main Review:**

I am leaning towards weak reject. The contribution, although very simple, seems to do well on the point completion task. However, I have concerns on the limited amount of experiments.

Insufficient experiments: My main issue with the work is that the experimentation is only on point completion. Chamfer is also used for many other application, e.g. ICP. If the argument is that the new metric is superior than Chamfer, then it needs to be shown on a few applications where Chamfer is usually the distance measure of choice. If the argument only applies to points completion, the scope of the problem seems too small. Alternatively, one could measure downstream metrics as well that uses point cloud matching, e.g. structured depth. Given how simple the new metric is, I feel that the experimentation has to be extended by a substantial amount.

Some other major feedbacks and questions:
- ln 241: why not using BCD as the loss?
- Instead of using the new metric, could we just use the product of CD and EMD (or the sqrt(CD) * sqrt(EMD))? Wouldn't we have the same behavior as the ones in figure 4?

Nit:
- Was the claim that the new metric is more robust substantiated anywhere?
- Figure 1: FPS -> farthest point sampling (its not a trivial abbreviation)
- ln 48. violently? better word choice please


**Time Spent Reviewing:**

1.5 - 3 hrs, didn't count exactly

---

> ### Author Response · Authors · 2021-08-10
> **Thank you for your valuable review**
>
> Thanks for your insightful questions and suggestions, and our answers are as below:
>
> ***Question 1: the experiments are only on point completion and can be extended to other applications.***
>
> Thanks for your suggestion, and we think it is a good idea to apply BCD in more applications. We will try it on more tasks in the experiments and expand the scope of the problem.
>
> ***Question 2: why not use BCD as the loss.***
>
> We tried to use BCD a training loss for a baseline PCN framework. The results of the different metrics when training with each of them are provided below:
>
> | Metric / Loss |   CD  | CD + EMD |  BCD  | BCD + CD-finetune |
> |:-------------:|:-----:|:--------:|:-----:|:-----------------:|
> |       **CD ($\downarrow$)**      |  9.78 |   10.70  | 11.21 |       10.56       |
> |       **F1 ($\uparrow$)**        | 0.326 |   0.310  | 0.338 |       0.341       |
> |      **EMD ($\downarrow$)**      |  6.80 |   3.97   |  4.94 |        4.92       |
> |      **BCD ($\downarrow$)**      | 0.553 |   0.537  | 0.518 |       0.518       |
>
> The F1 score is calculated based on CD following [16]. The training loss is indicated by the title of each column: "CD + EMD" denotes that the coarse shape is supervised by CD loss since EMD does not support a mismatched number of points, and the fine shape is supervised by EMD; "BCD" denotes a network trained with BCD and "BCD + CD-finetune" denotes that we train a model with BCD and fine-tune it with CD for ONE extra epoch.
> The results show that using BCD as training loss indeed promotes lower BCD results and higher F1 score, and it significantly reduces the EMD loss compared with the CD-trained baseline.
>
> However, we did not report the results in the submission for several reasons:
> 1) It is not fair to use a new metric for both training and evaluation since using it as the objective function will naturally lead to a lower value.
> 2) The improvement compared with CD + EMD trained baseline is marginal. We denote that the potential of BCD as a loss function is not entirely developed since it is not fully differentiable because of the definition of $n_x$ and $n_y$, as we also mentioned in the supplementary material. It remains our future work to develop a differentiable approximation of BCD to replace CD and EMD as the loss function, where we would expect it to produce even better results for CD and EMD metric themselves.
> However, this problem is currently out of the scope of this paper, and it does not prevent it from being a comprehensive metric at evaluation time.
>
> ***Question 3: could we use the product of CD and EMD (or sqrt(CD) * sqrt(EMD)) to get the same behavior.***
>
> Thanks for the suggestions, but the formulation has several drawbacks: first, the physical meaning is not clear and it cannot be viewed as a distance metric with the clean formulation. And then, this is not efficient and the time consumption is even heavier than pure EMD. But we will consider adding formulations like this that ensembles CD and EMD as an extra baseline.
>
> ***Question 4: where is the claim that the new metric is more robust substantiated in the paper.***
>
> Sorry for the confusion. By "robust" we mean the consistency of evaluation across different samples from different methods. We would modify the phrase here.
>
> ***Question 5: some representation issues.***
>
> Thanks for your suggestions, and we will polish the writing carefully.

---

> > ### Comment · Reviewer_ziDa · 2021-09-03
> > **Sorry for the last minute reply**
> >
> > Thank you very much for the extensive rebuttal. If I read it correctly, the authors mostly agree with my assessments (except for Q2). Hence I'll keep my ratings since there is no new information, and the co-reviewers have shared similar concerns. I acknowledge and I am satisfied with the Q2 reply.
> >
> > Should this work be rejected, I'd encourage the authors to add experiments on additional applications to show that BCD indeed can be a replacement for CD and EMD across the board, and not just for one application.

---

### Official Review · Reviewer_1bxc · 2021-07-16

**Rating:** 5
**Confidence:** 3

**Summary:**

This paper proposes an improved chamfer distance to evaluate the similarity of two point clouds.

**Limitations And Societal Impact:**

The authors address the potential limitations in the supplement material.

**Main Review:**

Originality: This work is an improved version of chamfer distance. Based on my knowledge, it is an original work and the paper has already explained the difference to other works clearly. However, I think the novelty is limited.

Quality: This work is technically sound and the claims are well supported. The method explains their strengths very well but does not mention the weakness that much. I think this paper is a work in progress.

Clarity: This paper is well-written and well organized.

Significance: This work is also medium significant.

**Time Spent Reviewing:**

5

---

> ### Author Response · Authors · 2021-08-10
> **Thank you for your valuable review**
>
> Thanks for your questions and suggestions, and our answers are as below:
>
> ***Question 1: the novelty is limited.***
>
> We discuss the problems with CD and EMD in the paper and develop a new metric that exactly addresses the issues of CD. We believe that the discussion on CD from this point of view is novel and would be helpful for many other works. What's more, evaluation metrics are essential for many tasks on point clouds since they directly affect the comparison among different methods. We hope that this work can draw more attention to this problem.
>
> ***Question 2: This work is technically sound and the claims are well supported. The method explains their strengths very well but does not mention the weakness that much.***
>
> Thanks for recognizing our work as technically sound with well-supported claims. The limitations of the work is discussed in our supplementary materials (Sec. D).

---

### Official Review · Reviewer_UpWV · 2021-07-16

**Rating:** 5
**Confidence:** 5

**Summary:**

The paper presents a new similarity measure named Balanced Chamfer Distance (BCD), which is derived from the original Chamfer Distance (CD) through an approximation of Taylor Expansion.

The authors propose a better sampling based on point query frequency to improve the overall performance for point cloud completion.

The authors also empirically demonstrate that BCD provides a more robust evaluation.

**Ethics Review Area:**

["I don’t know"]

**Limitations And Societal Impact:**

The major limitation of the paper is that the metric is limited for evaluation only.

**Main Review:**

**Originality**:
The proposed BCD makes a Tayler expansion to approximate the $x^2$ with the first order Tayler expansion $e^{x^2} \approx 1 + x^2$ ($x^2 \approx 1 - e^{x^2}$). Therefore, it bounds each point-level distance to a value between 0 and 1. Also, due to the exponential function, it potentially mitigates the outlier prediction when $x^2$ becomes large.

To further handle the imbalance ambiguity of CD, the authors propose to add one extra normalization term for each point-level distance by considering points’ surroundings, point-specific query frequency. Based on the frequency, the authors also designed a discriminator to guide point sampling to sample important points.

The reviewer considers the overall pipeline to be novel.

**Quality**: The paper is intuitive and straightforward to implement.
However, the major limitation the reviewer captures from the paper is that the BCD is only used during the test stage. As both CD and EMD are widely used for self-supervised learning for point-related problems, it seems to be quite unfortunate that the authors didn’t try to use BCD as the objective function to guide the learning tasks and report the numbers. This could be making it more useful for point cloud literature.

As the authors only use BCD for evaluation, the improvement made from the paper seems to mainly come from an empirical approach to conduct better sampling compared to FPS based on point query frequency. It is somewhat an incremental contribution.

**Clarity**: It is not clear on the details of the proposed model. For example, the authors should have mentioned the baseline architecture without the balanced design. It is not clear to the reviewer whether the authors built their method upon PCN, PCN++, TopNet, MSN or VRC. Therefore, it is difficult for the reviewer to evaluate whether the balanced design indeed improves the performance.

**Significance**: The authors propose a bounded version of Chamfer Distance, the Balanced Chamfer Distance, for distance measurement on point clouds. It is tackling important problems that are interesting to the point cloud literature.  However, its limitation is obvious that the metric is only limited for evaluation and can not be used for training as CD and EMD.

In the end, the method of the paper tends to ‘degrade’ to designing a better sampling based on point query frequency, which is unfortunately considered as an incremental contribution.

Therefore, the reviewer tends to give a borderline rejection given its current version as the work seems to be still an on-going work that remains to explore the training with BCD. The authors have attempted solving an important research problem, the reviewer encourages them to further move forward along the direction.

[1] A Point Set Generation Network for 3D Object Reconstruction from a Single Image


**Time Spent Reviewing:**

4

---

> ### Author Response · Authors · 2021-08-10
> **Thank you for your valuable review**
>
> Thank you for recognizing our work novel and tackling important researching problems, and thanks for your insightful suggestions and questions. Our answers are as below:
>
> ***Question 1: BCD is only used during the test stage and cannot be used for training as CD and EMD***
>
> Since BCD is an extension of CD, it can be used as a loss function. We train a baseline PCN framework with different loss functions and report the results as below:
>
> | Metric / Loss |   CD  | CD + EMD |  BCD  | BCD + CD-finetune |
> |:-------------:|:-----:|:--------:|:-----:|:-----------------:|
> |       **CD ($\downarrow$)**      |  9.78 |   10.70  | 11.21 |       10.56       |
> |       **F1 ($\uparrow$)**        | 0.326 |   0.310  | 0.338 |       0.341       |
> |      **EMD ($\downarrow$)**      |  6.80 |   3.97   |  4.94 |        4.92       |
> |      **BCD ($\downarrow$)**      | 0.553 |   0.537  | 0.518 |       0.518       |
>
> The F1 score is calculated based on CD following [16]. The training loss is indicated by the title of each column: "CD + EMD" denotes that the coarse shape is supervised by CD loss since EMD does not support a mismatched number of points, and the fine shape is supervised by EMD; "BCD" denotes a network trained with BCD and "BCD + CD-finetune" denotes that we train a model with BCD and fine-tune it with CD for ONE extra epoch.
> The results show that using BCD as training loss indeed promotes lower BCD results and higher F1 score, and it significantly reduces the EMD loss compared with the CD-trained baseline.
>
> However, we did not report the results in the submission for several reasons:
> 1) It is not fair to use a new metric for both training and evaluation since using it as the objective function will naturally lead to a lower value.
> 2) The improvement compared with CD + EMD trained baseline is marginal. We denote that the potential of BCD as a loss function is not entirely developed since it is not fully differentiable because of the definition of $n_x$ and $n_y$, as we also mentioned in the supplementary material. It remains our future work to develop a differentiable approximation of BCD to replace CD and EMD as the loss function, where we would expect it to produce even better results for CD and EMD metric themselves.
> However, this problem is currently out of the scope of this paper, and it does not prevent it from being a comprehensive metric at evaluation time.
>
> ***Question 2: it is not clear on the details of the proposed model and difficult to evaluate the performance improvement.***
>
> Sorry for the confusion. Our model is based on VRCNet [16] since it is the current SOTA method on the MVP dataset, and we will make this clear in the paper. The effect of the proposed balanced design and how it improves the performance from the baseline model is studied in the ablation study (Sec. 5.3, Table 3, and Figure 6).
>
> ***Question 3: the proposed method is considered to be an incremental contribution.***
>
> In this paper, we mainly focus on proposing a new metric to tackle the issues of CD and EMD and we select several typical approaches that reveal the problem of inconsistent evaluation.
> The proposed method is not the main focus of the paper, while it provides a solution and helps to give some examples for the visualization and statistical results in Sec. 5.1. The proposed module can also be applied to other frameworks with minor extra costs.
> What's more, when designing the module, our *key idea* is to use the nearest neighbor querying frequency as a supervision signal that indicates the matching of local density, which is closely coupled with the designation of our metric. We believe that this idea could also be useful for other frameworks.

---

> > ### Comment · Reviewer_UpWV · 2021-09-10
> > **Response to the authors**
> >
> > Hi,
> >
> > Thank you for the response.
> >
> > The reviewer is leaning to reject the paper and keep the original rating.
> >
> > Here the reviewer chooses to 'expose' the concerns during the internal discussion. Hopefully, this could help the authors improve the current submission.
> >
> > My main concern is that the paper is still an ongoing research project where its actual effectiveness remains unclear to me even after reading the rebuttal.
> >
> > The paper begins with a Taylor expansion for one approximation of Chamfer Distance (Equation 3) and further adds temperature and normalization (Equation 4). Later on, they further introduce a point discriminator for deciding point sampling privilege (Equation 5 and 6).
> >
> > It confuses the reviewer:
> >
> > 1) The reviewer mentioned the concern that ``the method of the paper tends to ‘degrade’ to designing a better sampling based on point query frequency, which is unfortunately considered as an incremental contribution`` The authors haven’t addressed the issue in the rebuttal. The contribution from each component in the pipeline of  ``CD → Equation 3 → Equation 4 → Equation 5 and 6’’ remains unclear to the reviewer.
> >
> > 2) If the main goal is to show that BCD is better compared to CD and EMD, the authors should give us strong evidence rather than put a plain number. It is an incremental contribution with a trivial modification --- ``normalizing’’ the CD with point query frequency.
> >
> > Besides, if their goal is to propose the BCD as an evaluation metric, the paper should go beyond the Point Cloud Completion and compare against CD and EMD in various point cloud tasks. Therefore, the current evaluation is quite limited.

---

### Official Review · Reviewer_b3Fh · 2021-07-18

**Rating:** 5
**Confidence:** 5

**Summary:**

This paper proposes a new evaluation metric for point cloud processing, namely BCD (balanced CD). The authors began by discussing the shortcomings of the commonly used CD and EMD losses, which may result in inconsistent evaluation, especially for imbalanced point clouds. By some simple modifications, the BCD can alleviate the issues of CD loss and report more reliable results which are consistent with the visualization. The authors then explored the architecture for point cloud completion, focusing on intermediate coarse prediction. Some modifications are specifically designed to solve the issues of the current completion framework. The proposed completion network achieves the best results under most assessment criteria (CD, EMD, BCD).

**Limitations And Societal Impact:**

See the main review.

**Main Review:**

1. The authors begin by doing a thorough study of CD and EMD losses, two widely used assessment criteria for point cloud processing. The authors emphasize two critical aspects, the data balance (point distribution) and quality (close to the underlying surface). By theoretical analysis and experimental demonstration, the authors show CD and EMD address just one component and neglect the other, resulting in the contradiction between these two measures. Moreover, these two metrics are unbounded.

2. The suggested BCD is derived by simply modifying the CD loss. More precisely, the suggested BCD differs from the original CD in the following ways: BCD introduces Nx, Ny to resolve the density imbalance issue; BCD realizes a bounded metric by using exponential distances rather than L2 distances. These modifications are simple and reasonable to some extend.

3. The authors also provide a detailed discussion for the point cloud completion task. Similar to the proposed BCD metric, some modifications of the standard completion network are proposed to address the imbalance prediction issue of the intermediate coarse prediction. The newly suggested network can attain SOTA performance across almost all assessment criteria.

In summary, the reviewer appreciates the paper’s in-depth analysis of both the evaluation metrics CD, EMD and the completion networks. In the following, I list some potential limitations of this work:

1. The newly introduced Nx and Ny seem only valid to process uniformly distributed ground truth point sets, which are also referred to in L45 as "balancing." However, if the ground truth is not uniformly distributed, I'm curious if BCD can still provide a consistent result with such a design. I think the authors should have a more detailed discussion of this situation.

2. The proposed BCD is only treated as an evaluation metric. Why not use it as the training metric to replace CD or EMD?

3. Although the issues of CD and EMD are examined in-depth, the proposed BCD is also based on CD with minor modifications. I think that such changes may not entirely address the mentioned issues.


**Time Spent Reviewing:**

2

---

> ### Author Response · Authors · 2021-08-10
> **Thank you for your valuable review**
>
> Thanks for your insightful questions and suggestions, and our answers are as below:
>
> ***Question 1: whether BCD can provide a consistent result when ground truth is not uniformly distributed.***
>
> Thanks for pointing this out. By using the word "balanced", we indeed assume that the ground truth point set is uniformly distributed because in this way, we can evaluate the quality of the output point cloud regarding the ground truth. If the ground truth is not uniformly distributed, e.g., more points lie in areas with larger curvature or at the edge, BCD would expect the other point set to capture the same density distribution to achieve a lower distance. So a more proper phrase could be *density-aware*, and we will double-check the paper to avoid confusion.
>
> For many tasks including point cloud completion, up-sampling, and denoising, perfect ground truth is usually provided for evaluation. But we admit that there are other scenarios where the similarity metric is expected to compare between the "intrinsic and continuous shaped" under the imperfect and discrete point clouds, but this is out of the scope of our paper.
>
> ***Question 2: why not use BCD as a training metric to replace CD or EMD.***
>
> We tried to use BCD a training loss in a PCN baseline model. The results of the different metrics when training with each of them are provided below:
>
> | Metric / Loss |   CD  | CD + EMD |  BCD  | BCD + CD-finetune |
> |:-------------:|:-----:|:--------:|:-----:|:-----------------:|
> |       **CD ($\downarrow$)**      |  9.78 |   10.70  | 11.21 |       10.56       |
> |       **F1 ($\uparrow$)**        | 0.326 |   0.310  | 0.338 |       0.341       |
> |      **EMD ($\downarrow$)**      |  6.80 |   3.97   |  4.94 |        4.92       |
> |      **BCD ($\downarrow$)**      | 0.553 |   0.537  | 0.518 |       0.518       |
>
> The F1 score is calculated based on CD following [16]. The training loss is indicated by the title of each column: "CD + EMD" denotes that the coarse shape is supervised by CD loss since EMD does not support a mismatched number of points, and the fine shape is supervised by EMD; "BCD" denotes a network trained with BCD and "BCD + CD-finetune" denotes that we train a model with BCD and fine-tune it with CD for ONE extra epoch.
> The results show that using BCD as training loss indeed promotes lower BCD results and higher F1 score, and it significantly reduces the EMD loss compared with the CD-trained baseline.
>
> However, we did not report the results in Table 1 for several reasons:
> 1) it is not fair to use a new metric for both training and evaluation since using it as the objective function will naturally lead to a lower value.
> 2) the improvement compared with CD + EMD trained baseline is marginal. We denote that the potential of BCD as a loss function is not entirely developed since it is not fully differentiable because of the definition of Nx and Ny, as we also mentioned in the supplementary material. It remains our future work to develop a differentiable approximation of BCD to replace CD and EMD as the loss function, where we would expect it to produce even better results for CD and EMD metric themselves. However, this issue is currently out of the scope of this paper, and it does not prevent it from being a comprehensive metric at evaluation time.
>
> ***Question 3: BCD is based on CD with minor modifications and may not entirely address the mentioned issues.***
>
> Since CD is a widely recognized distance and has been adopted as the evaluation criteria in many research papers, we do not aim to build a brand new metric that is entirely different from CD. Instead, we analyze the key problem in the formulation of CD that prevents it from properly detecting the imbalance issue and make modifications accordingly. In this way, the users could make a clear comparison between them.

---

### Decision · Program_Chairs · 2021-09-28

**Decision:**

Accept (Poster)

**Comment:**

The reviewers all agreed that this submission should be rejected. The reviewers find that the method lacks sufficient novelty and significance, and that the experimental results are not strong enough. In particular, the results show that the proposed method does not provide a significant improvement when used as a training objective, and the experiments are not comprehensive enough for the task of evaluation.

As a (very) minor side note, I also noticed that the paper claims that the EMD requires the point sets to have the same number of points, which certainly isn't true. The EMD is just the Wasserstein distance, which can be evaluated exactly between any two discrete distributions regardless of the (finite) number of points in the support.

**Consistency Experiment:**

NeurIPS has a long history of experimentation. In 2014, NeurIPS ran an experiment in which 10% of submissions were reviewed by two independent committees to quantify the randomness in the review process. This year, we repeated a variant of this experiment to see how the quality of the review process has changed over time.  This paper was part of the experiment and was therefore assigned to two committees (consisting of reviewers, an Area Chair, and a Senior Area Chair) that reached independent decisions.  If both committees made the same recommendation, this recommendation was followed. If a single committee recommended acceptance, the paper was accepted (with the exception of a few cases in which the other committee identified what we considered a fatal flaw, e.g., an error in a key result).

This copy’s committee reached the following decision: **Reject**

The other committee assigned to the paper recommended **Accept (Poster)**.  You can find the other set of reviews, along with any follow up discussion with the authors here:
https://openreview.net/forum?id=B46BjXrLidN